# Promotion of Healthy Habits in University Students: Literature Review

**DOI:** 10.3390/healthcare12100993

**Published:** 2024-05-11

**Authors:** Sara Puente-Hidalgo, Camino Prada-García, José Alberto Benítez-Andrades, Elena Fernández-Martínez

**Affiliations:** 1Department of Nursing and Physiotherapy, Universidad de León, Ponferrada Campus, 24001 Ponferrada, Spain; 2Department of Preventive Medicine and Public Health, University of Valladolid, 47005 Valladolid, Spain; cprada@saludcastillayleon.es; 3SALBIS Research Group, Department of Electrical, Systems and Automatics Engineering, Universidad de León, Campus de Vegazana, 24071 León, Spain; jbena@unileon.es; 4SALBIS Research Group, Department of Nursing and Physiotherapy, Universidad de León, Campus de Vegazana, 24071 León, Spain; elena.fernandez@unileon.es

**Keywords:** university student, intervention, physical activity, stress, resilience, academic burnout, prevention, promotion

## Abstract

The increase in responsibilities, together with the multiple challenges that students face in the university period, has a direct impact on their healthy lifestyles. This literature review describes the benefits of promoting healthy habits in college, highlighting the fundamental role of prevention and promotion. A systematic review was carried out following the PRISMA recommendations, searching for information in the WOS and Scopus databases. On the other hand, a search was carried out within the existing and available grey literature. The review focused on finding information about physical activity, nutrition, and stress (with an emphasis on resilience and academic burnout) in university students. This bibliographic review includes 32 articles and six web pages, containing information on the benefits of physical activity, healthy habits, and health prevention. The information collected in this study shows that university students are exposed to multiple changes during this period, increasing as the academic years progress. At that time, their habits worsen, with low adherence to the Mediterranean diet, low physical activity, and high levels of stress, specifically increasing cases of academic burnout. The establishment of healthy habits during the university period is necessary, observing an improvement in all the variables studied. Prevention has played a fundamental role.

## 1. Introduction

The objective of this literature review is to describe the benefits of promoting healthy habits at university, highlighting the fundamental role of prevention and anticipating the creation of routines that avoid the acquisition of unhealthy habits.

The information collected in this study shows that the increased responsibilities and multiple challenges faced by university students can affect healthy lifestyles, this being a key moment for adherence to healthy routines and habits that will remain in the future [1].

This literature review contains information on the benefits of physical activity, healthy habits, and health prevention.

In this context, this article has focused on three main aspects: physical activity, nutrition, and stress, including burnout and resilience.

Physical activity is essential for health, preventing numerous diseases and contributing to improved quality of life through multiple benefits at different levels, including psychological and physiological [2]. Physically inactive students have been shown to exhibit low intrinsic motivation, difficulty in emotion regulation, and alterations in certain dimensions of health-related quality of life [3]. However, participation in sports activities promotes social support and well-being [1]. Improved healthy lifestyle habits and increased physical activity have been observed with “reward” applications in students [4,5]. They are usually measured with two questionnaires with proven validity in university students: PAQ-AD and IPAQ-SF.

Eating habits during the university period undergo significant variations, decreasing the quality of the diet [6]. To assess the nutritional status of students, the KIDMED test was used, which measures the quality index of adherence to the Mediterranean diet [7,8,9,10], this being the dietary pattern to be implemented in students, since poor diet predisposes them to situations of obesity and the possible onset of diseases in adulthood [11,12].

The main interventions in this group: promotion of healthy foods in the main meals, together with daily physical activity. According to research by Cuberos et al., 2018 [13], those students with greater intrinsic motivation toward sports practice report healthier habits, so creating motivational climates in this area in the university community may be a fundamental strategy. González-Valero et al., 2009 [11] affirms that this motivational climate plays a fundamental role in physical activity, closely related to greater adherence to the Mediterranean diet. On the other hand, again guided by the research conducted by Cuberos et al., 2018 [13], a certain relationship of stress with nutritional problems has been observed, relating high levels of stress to a worse state of health, with less activity and worse diet quality, stress being another fundamental aspect to be addressed to avoid harmful habits.

Agah et al., 2021 [14], and Ozturk and Tezel, 2021 [15], show us in their research that university students face multiple stressors that can alter their mental health. This stress is especially intensified in the first year of university; specifically, the most stressful time is generated by the examination period [14,15].

Mental health problems in university students are very prevalent, as reported in their research Porru et al., 2021 [16], the association between “effort-reward imbalance” and “overcommitment” being important, as both are directly involved in the psychological distress of university students. The most prominent interventions to optimize academic performance and manage stress in university students are found in the research of Shinde et al., 2021 [17], on mindfulness training; and the research of Ozturk and Tezel, 2021 [15], on laughter yoga, observing that this discipline improves cortisol and stress levels in university students.

In direct relation to stress, we highlight two concepts: resilience and burnout. The first concept, resilience, is defined as healthy integration and adaptation over time in response to adversity and challenges [18]. Some college students lack the necessary skills to cope positively with the crises inherent to the college stage [19]. For this reason, mental resilience is key to student well-being and performance, and its measurement is important to support mental health and well-being [18,20].

To measure resilience, we have the resilience scale 14 [21], although there are also studies that endorse the resilience scale-10 in university students, self-determination and adaptability being the two key points considered as valid measures in this group in particular [19].

The proactive factors of resilience show perceived self-efficacy and the capacity for adaptive change, while reactive factors are associated with uncertainty, trauma, and enduring change experiences [22].

The second concept we highlight related to stress is burnout, recognized by the World Health Organization (WHO) as an occupational disease in May 2019. In the International Classification of Diseases ICD-11 of 2022, it includes this syndrome as a problem associated with employment or unemployment [23]. Burnout has traditionally been associated with work, but in recent years it has been shown that this syndrome is also increasing in academia, specifically in university students [24]. The number of studies is still scarce, which highlights the need for research in this group for a better understanding of this profile [25,26,27]. To measure academic burnout in university students, the use of the MBI-SS questionnaire is considered appropriate [24,25,28,29,30,31,32].

The use of the MBI-SS questionnaire reveals a high percentage of students with burnout symptoms [28], and signs of depressive symptoms were even observed in some of them, establishing a relationship between burnout and risk of depression [25,31].

Burnout in students has both individual and environmental explanations, and its prevention would require both organizational and individual interventions [25]. As for its associated factors, it has been observed that the frequent use of stimulant substances, together with alcohol and tobacco, increases the risk of burnout [27].

The design of academic burnout prevention programs in university students, with interventions focused on pedagogical and psychological support, is considered fundamental [27,32]. It has been observed that intervention based on the promotion of healthy lifestyle habits, focused on physical activity together with relaxation techniques, reduces stress. All activities related to mindfulness, such as yoga, are beneficial. A correct adherence to the Mediterranean diet and the exclusion of negative habits such as the consumption of alcohol and stimulant substances are also beneficial. It is essential to focus on primary prevention, avoiding high levels of stress, promoting healthy lifestyles, and recommending the reduction in triggering risk factors [33,34,35,36].

## 2. Materials and Methods

This review was based on the PRISMA methodology [37].

### 2.1. Review Questions

What healthy habits are necessary to promote in university students?What are the benefits of establishing healthy habits during college?

### 2.2. Eligibility Criteria

In order to guide the information search process, the PICO strategy [38] was used:Population: university students.Intervention: promoting healthy habits in university students.Comparison: studies showing improvements in students with habit change.O/Results: benefits of establishing healthy habits in a permanent way during the university period.

### 2.3. Sources of Information

In order to obtain the information, two search strategies were used: on the one hand, a search was carried out in the WOS and Scopus databases.

In relation to the consulted grey literature, specifically the website of the Ministry of Health and Consumer Affairs, the section on citizen—health promotion and prevention—benefits of physical activity was explored. The other web pages consulted addressed healthy habits: minervamedica.it and eu-osha.eu. In addition, pages related to health prevention were reviewed: prevention of occupational hazards—prlceoe.es (Ministry of Labor, Migration and Social Security), discapnet.es (prevention of occupational hazards), and dirdascalia.es (burnout as an occupational disease) [39,40,41].

### 2.4. Search

In order to focus the search for information in relation to the subject of the review, different key words were chosen according to the variables studied, having in common: “university student” and “intervention”.

Physical activity: “physical activity”, “scale”.Nutrition: “mediterranean diet”.Stress, resilience, and burnout: “stress”, “resilience”, “resilience scale”, and “academic burnout”.

To develop the search strategy, controlled language was used through Health Sciences Descriptors (DeCs), subsequently combining them with the Boolean operators “AND” and/or “OR”. Table 1 shows the database search strategy.

### 2.5. Data Extraction Process

The inclusion criteria were as follows:All studies were conducted between 2007 and 2023.The studies were in Spanish or English.The studies focused on university students.

The exclusion criteria were as follows:Duplicates.All studies that had a direct relationship or some kind of link to the COVID-19 pandemic. All studies related to the COVID-19 pandemic were discarded because all students’ habits and stress levels were altered at that time. The entire population, including university students, was exposed to previously unknown levels of stress and our routine was affected, so studies related to that event were not representative for this review.

### 2.6. Selection of Studies

In order to assess the quality of the selected articles, they were analyzed by means of a series of checklists developed by the Joanna Briggs Institute (JBI) [42,43,44,45].

In the present work, we made use of checklists created for the evaluation of articles with experimental and quasi-experimental design, cross-sectional studies, prospective studies, research articles, randomized controlled trials, and systematic reviews [46].

The articles were selected following the word frequency queries, highlighting the following: physical activity, nutrition, university students, and academic burnout.

As for coding, we were also guided by the following keywords: physical activity, nutrition, university students, and academic burnout.

### 2.7. Item Management

Mendeley was used as a bibliographic manager in order to organize the selected articles, allowing for the elimination of all duplicate studies, being used at the same time for the writing of citations and bibliographic references.

## 3. Results

With the search criteria indicated above, we found 82 articles and 12 web pages. Following the inclusion and exclusion criteria explained above, the final result was 32 articles and six web pages. Figure 1: Systematic review flowchart PRISMA.

Based on the information on authorship, research design, objectives, intervention performed, results, date of publication, country, and journal, the summary of the documents included in the review are shown in Table 2.

According to the typology of the resources obtained through the bibliographic search carried out with the criteria explained above, the studies found were cross-sectional studies, prospective studies, randomized controlled trials, and systematic reviews.

After a complete reading of all the documents, the information contained in them was grouped according to the study variables initially identified, which were as follows:

Physical activity: The selected studies indicate that a lack of physical activity hinders the regulation of emotions and affects the quality of life, thus being essential for health, preventing diseases, and obtaining benefits at psychological and physiological levels while able to measure them by means of the questionnaires PAQ-AD and IPAQ-SF [1,3,4,5,47]. Physical activity is the fundamental pillar of stress prevention in university students, promoting coping strategies and improvements at multiple levels.

Nutrition: It has been shown that, during the university period, eating habits undergo significant variations after assessment of nutritional status, highlighting the need to promote the Mediterranean diet, to avoid obesity and possible disease in adulthood. Interventions in this area focus on promoting a healthy diet together with the daily practice of physical activity. Finally, this highlights the link between poor diet and higher levels of stress [6,7,8,9,10,11,12,13]. Again, stress prevention is associated with a correct adherence to the Mediterranean diet, increasing this prevention with the help of physical activity. A good connection between nutrition and physical activity favors the improvement at multiple levels of the state of university students, improving stress levels and reducing academic burnout [48].

Stress: The selected studies show the increase in stress in university students especially in the first year, facing multiple factors that alter their mental health. Mindfulness and yoga may be the main interventions to try to decrease the level of stress [14,15,16,17].

Resilience: A lack of skills has been observed in university students who face crises inherent to the university period in a negative way, and resilience is a key factor in the well-being and performance of students. Therefore, it is essential to measure it and for this purpose we have the Resilience Scale-14 and the Resilience Scale-10 [18,19,20,22].Burnout: An increase in this pathology has been observed in university students, although the number of studies in this area is still scarce. The MBI-SS questionnaire is used for its measurement, showing a high percentage of students with burnout symptoms and even, in some of them, symptoms of depression. The design of academic burnout prevention programs is essential, with interventions focused on psychological and pedagogical support, implementation of healthy habits based on physical activity, correct adherence to the Mediterranean diet, and relaxation techniques [23,24,25,26,28,29,30,31,32,33,34,35,36,49].

## 4. Discussion

In this literature review, a total of 32 articles were selected in order to describe the benefits of promoting healthy lifestyle habits to university students.

As reflected by Boone et al. (2021) [1], in their study conducted at a US University, increased stress is a natural component expected in college life, with participation in extracurricular activities being a means to enhance the students’ well-being by promoting social support, considering this as an important form of stress prevention.

The results show that students who have a higher intrinsic motivation to participate in sports and perceive a climate of involvement in the task report healthier habits. This highlights the importance of creating motivational climates related to physical activity [11]. As referenced by Faílde-Garrido et al. (2021) [3], physically inactive students show lower intrinsic motivation, with worse emotional regulation and worsening in health-related quality of life. These results from this study can be useful when designing and implementing programs aimed at promoting health and physical activity, with special attention to physically inactive young populations. In this aspect, Cuberos et al. (2018) [13], in their study conducted on university students in Spain, corroborates these findings. Students who have greater intrinsic motivation to participate in sport and perceive a climate of involvement in the task, report healthier habits, coinciding with González-Valero et al. (2019) [11], who also conducted their studies on university students in Spain, re-emphasizing again the importance of creating motivational climates related to sport, considering physical activity as a healthy component and an aid to self-improvement learning as well. In this area, Lemola et al. (2021) [4], who conducted their studies in a UK University, and Reynolds et al. (2021) [5], who focused on students at a University of New Wales in Sydney, agree that mobile applications provide useful information and especially that exercise-incentive applications could increase physical activity levels. Both studies have reported satisfactory results, with improvements in adherence to physical activity through the use of incentive apps.

In all these studies focused on physical activity, satisfactory results have been reported with improvements in multiple aspects of students’ lives. This can be extrapolated to an international context since, being studies carried out in different parts of the world, the results coincide.

One of the most outstanding aspects in relation to nutrition in university students is that their diet is of low quality, with an intermediate-to-low adherence to the Mediterranean diet, with several authors agreeing on the need to establish nutritional education campaigns for university students [6,9,10]. This theory based on the need to establish educational campaigns makes sense when reviewing the article by Vega et al. (2014) [12], which performs an evaluation using the KIDMED questionnaire pre- and post-teaching in Spanish university students, based on health education, collecting significant variations after teaching about the Mediterranean diet. In this sense, Atencio-Osorio et al. (2020) [7] evaluated the psychometric properties of the KID-MED questionnaire in Colombian university students, demonstrating its reliability and validity. On the other hand, Badicu et al. (2018) [8] conducted a study in students from Romania and Spain using the KIDMED questionnaire and the PAQ-A, determining that students from both sites have low adherence to the Mediterranean diet and low performance in physical activity.

Another point to highlight in relation to stress in university students, as indicated by Porru et al. (2021) [16], in their study of Italian university students, is that one in three university students manifests severe psychological distress, showing the need for longitudinal studies to delve into the cause of this symptomatology and focusing on the prevention of severe psychological distress with the decrease in effort–reward imbalance and overcommitment.

As reported by Kajjimut et al. (2021) [26], who conduct their studies on university students in Africa; Dahlin and Runeson (2007) [25], in their study conducted on university students in Stockholm; Lane et al. (2020) [34], studying university students in Dublin; Merlo and Rippe (2021) [35] and Nteveros et al. (2020) [36],who focus on university students in Cyprus, university students experience some degree of burnout, which increases as the academic years progress. They agree that it is important to know the predictors of burnout and its relationship with life habits, sleep quality, and mental health, as the toxic effects of stress can affect the student’s cognition, learning, and empathy. This burnout has both individual and environmental explanations, and Dahlin and Runeson (2007) [25] and Merlo and Rippe (2021) [35] agree on the importance of prevention and of addressing early signs, highlighting the importance of a model to reduce risk factors focused on primary, secondary, and tertiary prevention. Kajjimut el al (2021) [26] highlights burnout prevention based on interventionist measures, and Ntveros et al. (2020) [36] focuses prevention on improving healthy habits in general.

In this context, Shinde et al. (2021) [17], who conducted their study on students at a Midwestern university, and Ozturk and Tezel (2021) [15], show how mindfulness training and laughter yoga reduce stress and stress-related symptoms in university students considering both preventive disciplines on stress.

Galán et al. (2021) [28], Ilic et al. (2017) [29], Portoghese et al. (2018) [24], Puranitee et al. (2019) [31], and Obregón et al. (2020) [30] agree on the validity of the MBI-SS scale to measure burnout in university students.

In direct relation to academic burnout, the importance of resilience is highlighted, identifying the attributes of individuals who seem to cope more adaptively with stressful situations and experience better overall well-being. These results could be useful in identifying students at risk for increased vulnerability to stress and burnout, in order to provide them with appropriate resources and support [20]. De la Fuente et al. (2021) [22], who conducted their study on Spanish university students, focus on resilience, positivity, and coping strategies as predictors of engagement in the context of burnout in university students, making proposals for therapeutic intervention for prevention in different student profiles, considering the work on resilience fundamental to withstand experiences of change. Jardim et al. (2021) [19], who studied Portuguese university students; and Sigley—Taylor et al. (2021) [18], who focused on Australian university students, highlight the importance of defining resilience and its implications in students.

Despite the disparity of interventions and recommendations, after analysis of the results, it was determined that academic burnout is significantly associated with resilience and optimism, highlighting the need to implement academic burnout prevention programs [32] based on health coaching, relaxation, and physical activity, together with cognitive behavioral therapy if necessary [14,33].

Likewise, prevention on burnout in university students and the importance of resilience in the occurrence of burnout can be contextualized internationally, since the reviewed studies have been conducted in multiple countries and to students from all over the world, reporting similar results.

### Limitations

One of the main limitations of the present work has been the lack of studies focused on university students, mainly on academic burnout.

In this context, and given that burnout is considered an occupational disease, its scope of study is focused more on adults, and the literature focused on university students is still scarce.

It is also worth noting as a limitation the lack of studies with results focused on long-term interventions; concrete improvements have been observed but reports over several months are still to be explored.

Likewise, and continuing in the context of stress, multiple studies have been found that relate stress to the COVID-19 pandemic, which should be discarded.

## 5. Conclusions

The aim of this literature review is to describe the benefits of promoting healthy habits during the university period, highlighting the fundamental role of prevention and promotion.

The information collected in this study shows that university students are exposed to multiple changes during this period, which increase as the academic years progress. At that time, their habits worsen, with low adherence to the Mediterranean diet, low physical activity, and high levels of stress, specifically increasing cases of academic burnout.

Regarding physical activity, it has been observed that daily sport levels favor the general wellbeing of students as well as offering social support. Hence the need for interventions based on the promotion of health and physical activity in this population group, reporting satisfactory results in the design and implementation of programs aimed at promoting health and physical activity, with improvements in multiple aspects and considering it fundamental in the prevention of stress and promotion of well-being.

In relation to nutrition, university students have a low adherence to the Mediterranean diet. An improvement in this domain is observed in students having received health education in university, reporting better adherence to the Mediterranean diet, this adherence being increased when combined with greater physical activity.

High levels of stress and the appearance of academic burnout, together with resilience, improved with interventions based on physical activity, health coaching, and relaxing activities. Again, prevention is a fundamental part, and cognitive behavioral therapy models have been observed to prevent stress in college students; severe psychological distress is diminished when the effort–reward imbalance and overcommitment are controlled. Physical activity improves stress and academic burnout; training programs decrease stress and improve life satisfaction. Multimodal programs focusing on relaxation, physical activity, and balneotherapy have shown short-term stress reduction. Coping mechanisms with work on resilience report positive results.

After reviewing all the literature, we can conclude by indicating that it is necessary to establish healthy habits in the university period, reporting multiple benefits with improvements in stress, favoring its reduction and achieving levels of satisfaction in multiple spheres of life, highlighting the fundamental role of prevention and promotion, helping to create routines that anticipate the establishment of harmful habits and the maintenance of healthy habits in the future.

## Figures and Tables

**Figure 1 healthcare-12-00993-f001:**
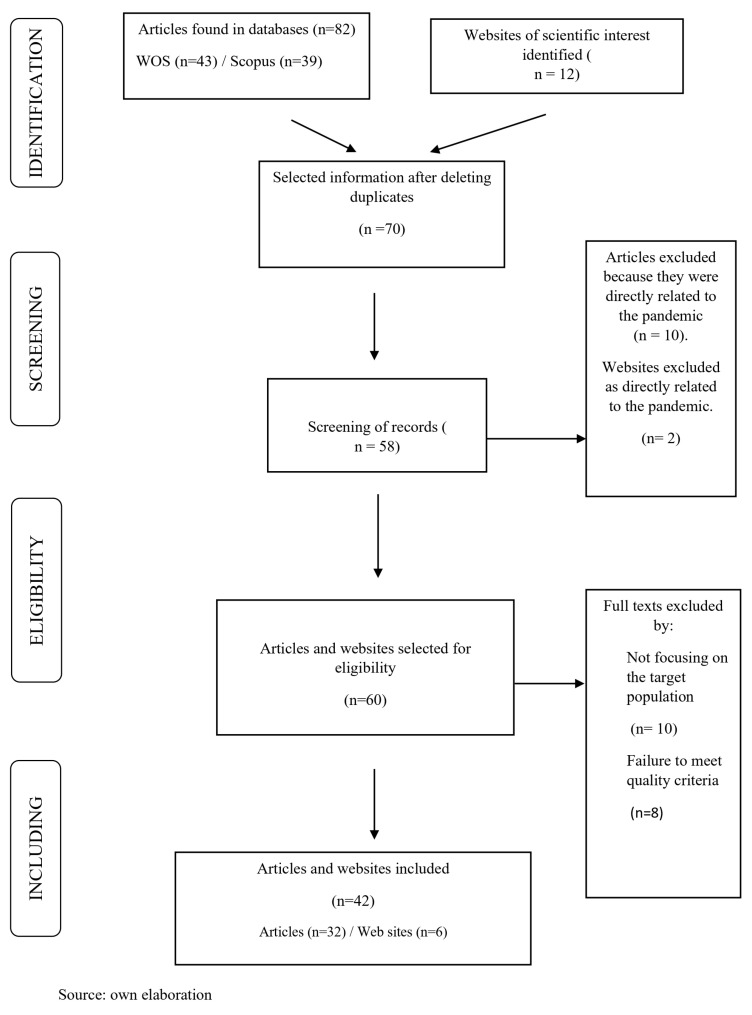
Literature review flowchart PRISMA. Results. With the search criteria, we found 82 articles and 12 web pages. Following the inclusion and exclusion criteria explained above, the result was 32 articles and 6 web pages. Source: own elaboration.

**Table 1 healthcare-12-00993-t001:** Database search strategy.

Data Base	Search Strategy	Search Description
**WOS**	“mediterranean diet” AND “university student”.	TS = (“mediterranean diet” AND “university student”)
	“physical activity” AND “university student”.	TS = (“physical activity” AND “university student”)
	“resilience scale” AND “university student”.	TS = (“resilience scale” AND “university student”)
	“academic burnout” OR “stress”.	TS = (“academic burnout” OR “stress”).
	“resilience” AND “university student”.	TS = (“resilience” AND “university student”).
**Scopus**	“mediterranean diet” AND “university student”.	TITLE-ABS-KEY (“mediterranean diet” AND “university student”)
	“physical activity” AND “university student”.	TITLE-ABS-KEY (“physical activity” AND “university student”)
	“resilience scale” AND “university student”.	TITLE-ABS-KEY (“resilience scale” AND “university student”)
	“academic burnout” OR “stress”.	TITLE-ABS-KEY (“academic burnout” OR “stress”).
	“resilience” AND “university student”.	TS = (“resilience” AND “university student”).

Source: own elaboration.

**Table 2 healthcare-12-00993-t002:** Presentation of selected documents. Summary of the documents included in the review, highlighting information on authorship, year, journal, and country of publication, as well as the main objectives, the intervention carried out, the results of interest, the research design, the researchers, quality and main prevention guidelines extracted by the study variables.

Variable	Authorship, Year, Journal and Country of Publication	Target	Intervention Performed	Results of Interest	Research Design	Search Engine	Quality (JBI)	Prevention
Physical activity	Boone et al., 2021—April 2021 [1]. *Journal American College Health*. USA.	To explore whether participation in extracurricular activities was associated with well-being and decreased suicidal ideation through perceived social support.	Recruited students (N = 583) from a southeastern US university completed a survey of self-reported measures.	Increased stress is an expected natural component of college life. Participation in extracurricular activities improves well-being by promoting social support.	Cross-sectional study	WOS	9/9	College extracurricular involvement as suicide prevention and wellness promotion.
Physical activity	Faílde-Garrido et al., 2021—April 2021 [3]. *Psychological Reports*. United Kingdom.	Comparative analysis between physically active and inactive college students in relation to motivational determinants, emotional self-regulation, physical fitness, health-related quality of life and other health parameters.	Comparative analysis in university students assessing motivational determinants, emotional self-regulation, physical fitness, health-related quality of life, and other health parameters.	Physically inactive participants showed lower scores on intrinsic motivation, emotional regulation, and on some dimensions of health-related quality of life.	Cross-sectional study	WOS	7/9	Useful results for the design and implementation of programs aimed at promoting health and physical activity, with special attention to physically inactive youth populations to improve quality of life.
Physical activity	Lemola et al., 2021—April 2021 [4]. *BMC Public Health*. United Kingdom.	It examined the impact of an “exercise rewards” mobile app on physical activity, subjective well-being, and sleep quality.	Three-month single-arm, open-label trial with staff at a university in the UK. They used the Sweatcoin app that converted their outdoor steps into virtual currencies used for the purchase of products available at the university’s retail outlets, using a marketplace within the app.	Mobile exercise incentive apps could increase physical activity levels, positive affect, life satisfaction ratings, and sleep quality in the short term.	Clinical trial	Scopus	11/13	
Physical activity	Reynolds et al., 2021—April 2021 [5]. *Health Promotion Journal Of Australia*. Australia.	A study of undergraduate student use of 10-day physical activity tracking apps at a University of New Wales, Sydney.	College students wore a Fitbit Zip and an iPhone, completing a modified Bouchard activity log (BAR) for 10 days in a free-living environment.	Mobile applications provide useful information for future pragmatic measurements of physical activity.	Cross-sectional study	WOS	7/9	
Nutrition	Atencio-Osorio et al., 2020—December 2020 [7]. *Nutrients*. Switzerland.	Evaluation of the psychometric properties of the KIDMED questionnaire in university students.	Cross-sectional study of a sample of 604 Colombian university students.	The study conducted in 604 university students on adherence to the Mediterranean diet provides reliability and validity.	Cross-sectional study	WOS	9/9	
Nutrition	Badicu et al., 2018—August 2019 [8]. *Behavioral Sciences*. Switzerland.	To show whether there are differences between Mediterranean diet and physical activity in university students in Romania and Spain using KIDMED and PAQ-A.	The study was conducted on a sample of 567 participants, 147 students from Romania, and 427 students from Spain. The KIDMED questionnaire and the PAQ-A questionnaire were used.	It was determined that there are no differences between adherence to the Mediterranean diet and the level of physical activity between the two countries.	Cross-sectional study	WOS	8/9	
Nutrition	Cabrera et al., 2015—September 2015 [9]. *Hospital Nutrition*. Spain.	Study to assess adherence to the Mediterranean diet among children and adolescents using the KIDMED test through a systematic review and meta-analysis.	The PubMed database was accessed until January 2014. Only cross-sectional studies evaluating children and youth were included. A random-effects model was considered.	The results obtained show a clear tendency toward the abandonment of the Mediterranean lifestyle.	Systematic review	Scopus	7/11	
Nutrition	Cervera Burriel et al., 2013- December 2012 [6]. *Hospital Nutrition*. Spain.	Cross-sectional study in university students on eating habits and nutritional assessment in a sample of 80 students.	Cross-sectional study of the habitual diet of an initial population of 105 university students of the Faculty of Nursing of Albacete (University of Castilla la Mancha) Spain.	The diet of the university population studied is of low quality, with an intermediate–low adherence to the Mediterranean diet, requiring changes for a healthier dietary pattern.	Cross-sectional study	Scopus	7/9	
Nutrition and physical activity	Cuberos et al., 2018—March 2018 [13]. *Sustainability*. Switzerland.	The objective was to establish and verify an explanatory model of motivational climate in sport that considers other possible influential variables related to health in 490 students in Spain.	Perceived motivational climate in sport was assessed using the 33-item PMC SQ-2 questionnaire. Alcohol consumption was assessed using the 10-item AUDIT test. Problematic video game use was assessed using the 17-item QERV questionnaire. Physical activity levels were assessed using the 10-item PAQ-A questionnaire. Adherence to the Mediterranean diet was assessed using the 16-item KIDMED questionnaire.	The findings imply that students who have greater intrinsic motivation to participate in sport and perceive a climate of task involvement report healthier habits.	Cross-sectional study	Scopus	8/9	Encourage healthy habits by promoting greater intrinsic motivation to participate in sports.
Nutrition and physical activity	González-Valero et al., 2019—April 2019 [11]. *Behavioral Sciences*. Switzerland.	The aim of the study was to determine the association between motivational climate, adherence to the Mediterranean diet, and the practice of physical activity in future physical activity teachers.	The sample consisted of 775 university students in Spain. Motivational climate was assessed using the PMCSQ-2 questionnaire. Physical activity levels were assessed using the PAQ-A questionnaire. The level of adherence to the Mediterranean diet was assessed using the KIDMED questionnaire.	The healthy and self-improvement component of physical activity promotes a process and learning-centered orientation. The competitive component is key to product-focused motivation and social recognition. Ego climate is related to high adherence to the Mediterranean diet.	Cross-sectional study	WOS	9/9	Physical activity as a way to promote healthy and self-improvement components focusing on process and learning.
Nutrition	Manzanera and Vega, 2014—May 2014 [10]. *Hospital Nutrition*. Spain.	To determine adherence to the Mediterranean diet in a university population of future teachers and to analyze various factors that could condition their nutritional quality.	Distribution of the KIDMED test to a sample of 212 Spanish university students.	It was established that 75.5% needed to improve their adherence to the Mediterranean diet.	Cross-sectional study	Scopus	8/9	
Nutrition	Vega et al., 2014—July 2014 [12]. *Hospital Nutrition*. Spain.	To determine the variations that teaching nutrition/food subjects can achieve both in knowledge and adherence to the Mediterranean diet in a university population of future nurses and teachers and to analyze various factors that could improve their nutritional quality.	Pre- and post-teaching distribution of a questionnaire and the KIDMED test to a sample of 399 Spanish university students.	There are significant variations after teaching about the Mediterranean diet. Despite the progress made, they would need to improve their adherence to the Mediterranean diet and it would be necessary to increase their Mediterranean dietary habits.	Cross-sectional study	Scopus	9/9	
Stress	Agah et al., 2021—March 2021 [14]. *Current Psychology*. Switzerland	This study explored the problem and factors associated with test-induced stress management among college students using the FEAR model of cognitive behavioral intervention.	The study employed a randomized pretest–posttest setting, with a total of 159 students from Nigeria participating. Recruited participants were assigned to two comparison groups using the sequence assignment method.	The study concludes that the cognitive behavioral therapy model is effective in reducing exam-induced stress in university students.	Cross-sectional study	Scopus	8/9	Cognitive behavioral therapy models prevent test-induced stress in college students.
Stress and physical activity	Ozturk and Tezel, 2021 —February 2021 [15]. *International Journal Of Nursing Practice*. Australia.	The aim of this study is to evaluate the effect of laughter yoga on mental symptoms and cortisol levels in nursing students.	A total of 75 healthy college students were assigned to the intervention and control groups. The questionnaire was administered to both groups before session 1 and after session 8. Saliva samples were taken from the students to measure their cortisol levels before and after each session.	Laughter yoga can provide an effective means to help first-year nursing students cope with stress and reduce mental health-related symptomatology.	Randomized controlled trial	WOS	11/13	Laughter yoga prevents stress and reduces symptoms related to mental health.
Stress	Porru et al., 2021—March 2021 [16]. *Journal Of Affective Disorders.* Netherlands.	This study explores mental health among college students, the association between effort–reward imbalance (ERI), overcommitment, and mental health, and the extent to which ERI and overcommitment explain gender differences in mental health.	Cross-sectional data from 4760 Italian university students were analyzed. The Kessler Psychological Distress Scale-10 and the ERI—Student Questionnaire were used.	One in three students reported severe psychological distress. Decreased ERI and overcommitment may be beneficial in preventing psychological distress among college students and may reduce gender differences in psychological distress.	Cross-sectional study	WOS	8/9	Prevention of severe psychological distress by reducing effort–reward imbalance and overcommitment.
Stress	Shinde et al., 2021—May 2021 [17]. *Accounting Education*. United Kingdom.	The aim of this study is to evaluate the effect of mindfulness training on stress, mindfulness, and life satisfaction in accounting students.	Randomized control trial. A mindfulness training program called Daata Meditation (DM) was used on a group of 88 students at a Midwestern university compared to students in a control group.	Students who went through a mindfulness training program called Daata Meditation (DM) were compared with students in a control group. Students who went through mindfulness training showed reduced stress, increased mindfulness, and greater satisfaction with life.	Randomized control trial	Scopus	12/13	Prevention of stress with a mindfulness training program observing a decrease in stress with increased life satisfaction.
Resilience and burnout	de la Fuente et al., 2021—February 2021 [22]. *Frontiers in Psychiatry*. Switzerland.	The objective was to establish a model with linear, associative, and predictive relationships to identify needs and make proposals for therapeutic intervention in different student profiles.	A total of 1126 Spanish students participated in the study. The Connor–Davidson Resilience Scale, CD—RISC, was used to measure resilience. The Positivity Scale was used to measure positivity. The original version of the EEC scale, validated for university students, was used to measure coping strategies. The MBI scale was used to assess burnout.	Proactive resilience factors reflect a perception of self-efficacy and the ability to change adaptively. Reactive factors of resilience are generally associated with enduring experiences of change, uncertainty or trauma, and the positive relationship with engagement factors.	Cross-sectional study	WOS	8/9	Reactive factors of resilience make it possible to withstand experiences of change.
Resilience	Fullerton et al., 2021—February 2021 [20]. *PLOS ONE.* USA.	Model of integrative resilience process in an academic context: resilience resources, coping strategies, and positive adaptation.	Thirty-six Australian undergraduate psychology students participated in exchange for academic course credit.	This study provides empirical support for existing theories of resilience, shedding light on relationships between individual differences and coping mechanisms associated with resilience and positive outcomes.	Cross-sectional study	WOS	8/9	Resilience-based coping mechanisms report positive results.
Resilience	Jardim et al., 2021—February 2021 [19]. *Education Sciences*. Switzerland.	Develop and validate a scale to assess student resilience in the face of adversity.	The scale was administered to a sample of 2030 Portuguese university students.	The obtained results pointed to a factorial structure composed of two factors called “self-determination” and “adaptability”, which showed good internal consistency. The scale proved to be a valid measure for assessing resilience among the university population.	Cross-sectional study	Scopus	7/9	
Resilience	Sigley-Taylor et al., 2021—April 2021 [18]. *Psychology in the Schools*. USA.	This study explored the relationship between a subjective and objective measure of resilience and the respective predictability of measures of psychophysical well-being in 282 Australian university students.	Two measures were used to quantify resilience levels in a sample of adolescents.	Highlights the importance of defining resilience and the implications for measurement in students.	Cross-sectional study	Scopus	8/9	
Burnout	Dahlin and Runeson, 2007—April 2007 [25]. *BMC Medical Education*. United Kingdom.	The aim of this prospective study of medical students was to examine clinically significant psychiatric morbidity and burnout in the third year of medical school, considering personality and study conditions measured in the first year.	Questionnaires were sent to 127 first-year medical students at a university in Stockholm, who were then followed up in the third year of the faculty, with 81 students in this course participating in a diagnostic interview. Personality and performance-based self-esteem were assessed in the first year. Study conditions, burnout, depression in the first and third year. Diagnostic interviews were used in the third year to assess psychiatric morbidity. Burnout in the third year was defined by cluster analysis. Logistic regressions were used to identify predictors of high professional burnout and psychiatric morbidity, controlled for gender.	Morbidity is common, but few seek help. Burnout has both individual and environmental explanations and organizational and individual interventions may be necessary to prevent it. It may be important to address early signs of depressive symptoms in medical students. Students should be encouraged to seek help and adequate facilities should be available.	Prospective study based on interviews	WOS	8/9	Prevention-based approach to early depressive symptoms in college students.
Burnout	Galán et al., 2011—March 2021 [28]. *International Archives Of Occupational and Environmental Health*. USA.	The first objective of this study was to investigate the prevalence of burnout risk in medical students using the MBI-SS questionnaire. The second objective was to investigate the association between gender and burnout subscales.	A cross-sectional study was conducted in a sample of 270 Spanish medical students, 176 third-year and 94 sixth-year, using the MBI—SS questionnaire.	The MBI-SS overcame the difficulties encountered when students have little or no contact with patients. The findings show that the risk of burnout prevalence doubled from the third year to the sixth year.	Cross-sectional study	WOS	9/9	
Burnout	Ilic et al., 2017—July 2017 [29]. *Behavioral Medicine*. United Kingdom.	The aim of this study was to assess the dimensionality of the MBI-SS in a sample of Serbian medical students.	The MBI-SS questionnaire was administered to a sample of 760 undergraduate medical students from Serbia.	The Serbian version of the MBI-SS represents a valid and reliable instrument in the Serbian sample of medical students.	Cross-sectional study	WOS	8/9	
Burnout	Kajjimu et al., 2021—January 2021 [26]. *Advances in Medical Education and Practice*. United Kingdom.	To determine the prevalence of burnout among undergraduate medical students in Africa, as assessed by the MBI-SS questionnaire, as well as factors associated with the development of burnout among students pursuing the Bachelor of Medicine and Bachelor of Surgery degrees.	A single-center, cross-sectional, online survey was conducted among students. Burnout was assessed using the MBI-SS tool.	More than half of the medical students surveyed experience some degree of burnout. Preventive and interventional measures should be considered in the development of the medical curriculum.	Descriptive cross-sectional study	WOS	8/9	Burnout prevention based on interventionist measures.
Stress	Kus et al., 2021—February 2021 [33]. *Complementary Medicine Research*. Switzerland.	The effectiveness of a one-week multimodal stress reduction program is investigated.	The intervention consisted of health coaching, relaxation, physical activity, and balneotherapy elements. Individuals were randomly assigned to the control group and the intervention group, collecting data before and after the intervention and after one, three, and six months.	The main outcome was the change in stress six months after the intervention. Other outcomes were well-being and health status. The results indicate that even a short-term multimodal stress reduction program appears to establish a positive trend toward less perceived and chronic stress.	Randomized controlled trial	WOS	12/13	Stress prevention through a multimodal program: relaxation, physical activity, and balneotherapy, showing short-term stress reduction.
Stress	Lane et al., 2020—December 2020 [34]. *BMI Open*. United Kingdom.	The main objective was to compare objective and subjective levels of stress in final-year medical students and to explore their perspectives on the factors they considered relevant to their well-being.	Participants in this study were senior Dublin undergraduate medical students, N = 240. Teaching follows a modular curriculum, the survey was conducted in week 5 of the 6-week Psychiatry Module, measuring stress levels in the 4 weeks prior, when students participated in clinical practicum and ongoing formative and summative assessments, and at least 10 days before the final modular assessment.	Medical students experience high levels of psychological distress.	Descriptive cross-sectional study.	WOS	9/9	
Burnout	Merlo and Rippe, 2021—April 2021 [35]. *American Journal Of Lifestyle Medicine*. USA.	Explores the process of burnout, including the historical context, international definitions, symptoms, and the imprecision of clinical diagnosis.	This article explores burnout, historical context, definitions, symptoms. Systemic etiological aspects and psychological underpinnings are explored, including personal vulnerabilities of physicians that contribute to burnout.	A prevention model is proposed for the reduction in risk factors, focusing on primordial, primary, secondary, and tertiary prevention.	Systematic review	WOS	9/11	Prevention of burnout through a model to reduce risk factors focused on primary, secondary, and tertiary prevention.
Burnout	Nteveros et al., 2020—November 2020 [36]. *PLOS ONE*. USA.	The main objective is focused on estimating the prevalence of burnout among all medical students at the Faculty of Medicine, University of Cyprus. The secondary objectives were to know the predictors of burnout and its relationship with life habits, sleep quality, and mental health.	An anonymous questionnaire was administered to 189 medical students at the University of Cyprus. It included demographic and lifestyle characteristics. Sleep quality was assessed using the Pittsburgh Sleep Quality Index. Mental health was assessed through the mental health (MH) domain of the 36-item Short Form Health Survey (SF-36). Burnout was measured with the MBI–SS.	As the academic year progresses, there are more students with burnout. Students with burnout were more likely to have poor sleep quality and poorer mental health. Alcohol consumers had more symptoms of cynicism and less feelings of efficacy than non-alcohol consumers. Burnout is frequent and increases during the clinical years. The students experience poorer sleep and mental health and may use alcohol as a coping mechanism. The implementation of burnout prevention strategies can be beneficial.	Cross-sectional study	WOS	8/9	Prevention of academic burnout through habit improvement.
Burnout	Portoghese et al., 2018—November 2018 [24]. *Frontiers in Psychology*. Switzerland.	The aim is to analyze the factorial validity, invariance, and latent profiles of the Italian version of the MBI-SS in a sample of university students.	A total of 7757 Italian university students participated. The Italian version of the MBI-SS was translated according to the Brislin procedure. In a second stage, two native English-speaking authors translated it.	The MBI-SS is valid and reliable and represents a robust instrument for the measurement of burnout among Italian-speaking university students.	Cross-sectional study	WOS	8/9	
Burnout	Puranitee et al., 2019—November 2019 [31]. *International Journal Of Medical Education*. United Kingdom.	To examine the psychometric properties of the Thai version of the MBI-SS and to determine the frequency of burnout and the correlation between burnout and associated factors in undergraduate medical students.	A cross-sectional study was conducted in Thai undergraduate medical students N = 545. The MBI-SS was translated into Thai and tested for internal consistency using Cronbach’s alpha coefficient.	The Thai version of the MBI-SS had adequate psychometric properties among Thai medical students and can be used to assess burnout. Burnout was associated with the risk of depression.	Cross-sectional study	WOS	8/9	
Burnout	Obregon et al., 2020—October 2020 [30]. *BMC Medical Education*. United Kingdom.	The objectives of this study were to assess burnout using the MBI-SS, identify factors that may predict burnout, and evaluate the effectiveness of wellness initiatives to reduce burnout in college students.	The MBI-SS was administered to all US undergraduate medical students. Factor analysis and internal consistency of the MBI-SS were evaluated. Mean MSBI-SS subscale scores were calculated for burnout, cynicism, emotional exhaustion, and academic efficacy.	Burnout in medical students was validated using the MBI-SS.	Cross-sectional study	Scopus	8/9	
Resilience and burnout	Vizoso-Gómez and Arias-Gundín, 2018—March 2018 [32]. *European Journal Of Education and Psychology*. Spain.	The relationship between resilience, dispositional optimism, and the dimensions that constitute academic burnout (emotional exhaustion, cynicism, and low efficacy) was analyzed in 463 students of the University of León.	A sample of 463 university students from the University of León, Spain, was evaluated. The following instruments were used: MBI-SS Resilience Scale 10 LOT-R	The results reveal that academic burnout is significantly related to resilience and optimism. The implications of the results obtained for the design of academic burnout prevention programs in university students are considered.	Cross-sectional study	Scopus	9/9	Prevention of academic burnout through programs that work on resilience and optimism.

Source: own elaboration.

## Data Availability

Not applicable.

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
