# Peer review of "Promotion of Healthy Habits in University Students: Literature Review"

_healthcare, 2024, doi:10.3390/healthcare12100993_

Round 1
Reviewer 1 Report
Comments and Suggestions for Authors
The text describes a systematic review focusing on the promotion of healthy habits among college students. The study was conducted in adherence to PRISMA recommendations, with extensive research across databases and grey literature. The findings, encompassing 32 articles and six web pages, indicate that university students experience increasing challenges with unhealthy habits as the academic years progress, leading to low adherence to the Mediterranean diet, low physical activity, and high stress levels, including academic burnout. The review underscores the imperative of establishing healthy habits during university years, with prevention emerging as a key driver in improving all the variables under study.
The aim of the study is clear; however, the research question "Is it necessary to promote healthy habits in university students?" sounds too obvious. It could be changed into a qualitative research question (what/how, etc.).
It was not sufficiently described how the Authors dealt with the "All studies that had a direct relationship or some kind of link to the Covid-19 pandemic" exclusion criteria. It could be explained.
Another weak point concerns the search procedure: the keywords in the two databases are different (Scopus can index texts in Web of Science, so maybe they could be the same).
Also, Authors should describe how they methodologically deal with grey literature.
The results were not analysed; they were just presented in the table. Word Frequency Queries or coding could be helpful techniques.
The results could be discussed in international context.
What concerns technical aspects, there are gaps between lines, such as 39-40 and 41-42 in the Introduction. They appear in the whole text and should be removed.
In the line 127, there is probably an unfinished sentence ("- Comparison - "). The text should be carefully revised before its submission.
I kindly encourage Authors to explore similar techniques employed in other articles. Familiarizing with related methodologies can enhance the quality and depth of research findings, ultimately contributing to the scholarly discourse in the field.
Regarding all the issues, at this stage I recommend rejection. However, after significant improvements, the text could be resubmitted.
Author Response
Please see the attached file. Thank you Please see the attached file. Thank you

Reviewer 2 Report
Comments and Suggestions for Authors
Dear authors, thank you for the opportunity to get acquainted with your interesting research.
Strengths of the study include the wide coverage of scientific publications on physical activity, nutrition and stress among university students.
The introduction contains detailed information about the relevance of the study for science and practice. It reflects the purpose of the study.
The method is described in detail; a sufficient number of publications have been selected to enable systematic analysis. Various search databases were used.
To illustrate the results of the study, the authors made a table with the main content of the articles.
It should be noted that the columns with characteristics for analysis are made very general, which does not allow detailed knowledge of the results regarding the purpose of the study.
Perhaps the authors should consider a more detailed analysis scheme, including what recommendations for prevention were in the articles or were already being put into practice (since prevention is stated as the purpose of the systematic review).
It may make sense to divide all the studies into the three groups that the authors talk about (physical activity, nutrition and stress), and present the studies in these three groups in the table.
The results discussion section also requires improvement. We need to expand it and add analysis. Now this is presented very briefly and superficially, which does not reveal the purpose of the study.
The conclusions are very general and do not fully outline the main results. Requires detail.
In connection with the above, the article requires significant revision.
Best wishes, reviewer
Reviewer 3 Report
Comments and Suggestions for Authors
Based on a review of research and literature, the authors...

Round 2
Reviewer 1 Report
Comments and Suggestions for Authors
Dear Authors,
Thank you for significantly improving the article. Your efforts have made it more transparent and polished. However, a few areas could benefit from further refinement.
Firstly, in line 25, could you please clarify what "6 web pages" means? For example, are we referring to six web pages containing information about...?
Secondly, in lines 60-75, consider a more formal style to enhance the credibility of the research. Instead of phrases like "this study found", perhaps phrases such as "According to the research of..." or "XYZ states that..." would be more appropriate.
While the Introduction provides a comprehensive overview of the topic, it may sound a bit too academic for some readers. The article's Introduction could benefit from a funnel structure, gradually narrowing down from general information to specific details about the research (objectives, methodology, and critical findings would help guide the reader smoothly into the study's content). This structured approach would enhance the clarity and coherence of the Introduction, ensuring that readers are adequately prepared for the in-depth exploration of the research presented in the article.
In line 136, since there are no subjects for comparison, would it be possible to find other research studies to provide a basis for comparison? This could add depth to the analysis and strengthen the conclusions drawn.
Lastly, in line 211, while you have identified the lack of studies focused on university students as a limitation, are there any other limitations that could be explored further? These could be integrated into the discussion more seamlessly rather than being presented as a separate point.
Overall, the article has made significant progress, and addressing these points would further enhance its quality and readability. Keep up the good work!
Reviewer 2 Report
Comments and Suggestions for Authors
Dear authors,
Thanks for the additions and corrections!
The article may be recommended for publication.
Best wishes, reviewer
